# Health professionals' perceptions on local production and bioequivalence study of generic medicines: A cross-sectional survey of physicians and pharmacy professionals in Addis Ababa, Ethiopia

Muluken Nigatu Selam[1], Abrham Temesgen Mengstu[1], Atalay Mulu Fentie[2], Gebremedhin Beedemariam Gebretekle[1,3,4], Eskinder Eshetu Ali[1], Anteneh Belete[1]*

**1** Department of Pharmaceutics and Social Pharmacy, School of Pharmacy, College of Health Sciences, Addis Ababa University, Addis Ababa, Ethiopia, **2** Department of Pharmacology and Clinical Pharmacy, School of Pharmacy, College of Health Sciences, Addis Ababa University, Addis Ababa, Ethiopia, **3** Institute of Health Policy Management and Evaluation, University of Toronto, Toronto, Ontario, Canada, **4** Toronto Health Economics and Technology Assessment (THETA) Collaborative, University Health Network, Toronto, Ontario, Canada

* anteneh.belete@aau.edu.et

## Abstract

Local production of generic medicines in developing countries has a critical role to meet public health needs by ensuring the availability of essential medicines and providing patients' relief from the burden of unaffordable medical bills. Compliance with bioequivalence (BE) requirements increase the quality and competitiveness of generic drugs regardless of the source. In this regard, a regional BE center has been established in Addis Ababa, Ethiopia to serve the needs of Ethiopia and neighbouring countries. The present study aimed to assess the knowledge and perceptions of health professionals working in Addis Ababa regarding local production and BE studies of generic medicines. A cross-sectional survey was employed and physician participants working at public hospitals and pharmacists from various practice settings were selected using convenient sampling technique. Data was collected using self-administered structured questionnaire. Descriptive statistics was used to summarize the data and multinomial logistic regression analyses was used to assess predictors of health professionals' perception towards the source of drugs. Statistically significant association was declared at p-value < 0.05. A total of 416 participants responded and 272 (65.4%) of them were male. Nearly half of the study participants (n = 194) preferred the imported products. Compared to physicians, participants with diploma (AOR = 0.40; 95%CI: 0.18–0.91, p = 0.028) and bachelor degree and above holders (AOR = 0.32; 95%CI: 0.15–0.68, p = 0.003) in pharmacy were more likely to prefer locally produced products. Participants who practiced in pharmaceutical industries (AOR = 0.40, 95%CI: 0.22–0.77, *p = 0.006*) preferred locally manufactured products as compared to those practicing in the hospital. While a majority (321, 77.2%) believed in the advantages of doing BE studies locally, only 106 (25.5%) recognized that local pharmaceutical manufacturers did not implement BE studies for their generic products and lack of enforcement by the national regulatory body

**Data Availability Statement:** All the minimum data set is included in the paper and as Supporting Information.

**Funding:** This research was financially supported by a research grant from the Addis Ababa University. However, the funder had no role in study design, data collection and analysis, decision to publish, or preparation of the manuscript.

**Competing interests:** The authors have declared that no competing interests exist.

was raised as a reason for not conducting BE studies by most of the participants (67.9%). The present study revealed a modest preference by physicians and pharmacy professionals towards locally produced products. Majority of participants supported the idea of doing BE studies locally. However, manufacturers and regulators should devise ways to increase health professionals' confidence in local products. Strengthening local BE study capacity is also highly recommended.

## Introduction

In recent years, the cost of pharmaceuticals has been escalating globally [1]. The situation is even worse in most sub-Saharan African countries that import up to 90% of their annual medicine requirements. In the context of these countries, the introduction of generic products has been found to be a successful strategy for lowering the cost and improve accessibility of medicines [2, 3]. The success and sustainability of generic medicine policies is partly influenced by countries' local medicine production capacities [4–6]. In fact, policies encouraging the production of generic medicines have been found to reduce prices by up to 89% [7, 8].

Since generic medicines are multisource products, their "interchangeability" with innovator products should be established by carrying out "*in vivo* equivalence" or "bioequivalence" (BE) studies [2, 9]. That is why many countries have developed legislations that evaluate, among others, the safety and efficacy of generic products before granting market authorization. This is typically established by conducting BE studies as per international guidelines [10–12]. The BE studies are generally done to evaluate whether the same molar dose of the alternative products administered under similar conditions have significant differences in terms of the rate and extent to which the active ingredient becomes available at the site of drug action [13].

Compliance with BE requirements has led to an increase in the quality and competitiveness of generic drugs regardless of the source. It has also helped to increase the confidence of healthcare providers and consumers with regard to their safety and efficacy [10]. In contrast, the absence of proof of BE studies or any concern in the quality of the available generic products undermine patients' and healthcare providers' confidence in these products [14, 15]. Lack of knowledge about BE and regulation of generics also affect healthcare providers' attitudes towards generic substitution [16]. The belief that "the more expensive the product, the more effective" is shared by quite a few physicians and pharmacists which leads to the consideration of low price generic products as low quality [17]. Thus, increased transparency regarding BE data and advocacy activities on BE of generics may help offset some of the concerns and enable physicians and pharmacists to make better choices [18]. This requires an understanding of the level of knowledge and perceptions of health professionals.

In Ethiopia, there are a handful of local pharmaceutical companies manufacturing generic medicines covering about 15% of the domestic demand [6, 19]. The rest of the demand is fulfilled by imported medicines from various countries, mainly from India and China [19, 20]. The Ethiopian Food and Drug Authority (EFDA) is the regulatory body responsible for approving new pharmaceuticals and generic medicines after checking their conformity to appropriate standards. Moreover, a regional bioequivalence center has been established to serve the needs of Ethiopia and neighbouring countries. However, there is little known about the perceptions of health professionals regarding BE studies. Therefore, this study aimed to evaluate physicians' and pharmacy professionals' knowledge and perceptions regarding local production and BE studies of generic drugs, and the barriers and perceived advantages of conducting local BE studies in Ethiopia.

## Methods

### Study area and participants

This study was conducted in Addis Ababa, Ethiopia among physicians working in public hospitals and pharmacy professionals working in public hospitals, pharmaceutical industries, community pharmacies, Ethiopian pharmaceutical supply services, EFDA, and Ministry of Health.

### Study design

A multicenter cross-sectional study design was used to collect data from physicians and pharmacy professionals.

### Sample size and sampling technique

The sample size was determined using the single population proportion formula with the following assumptions: proportion of participants who prefer local products = 50% (it was chosen as there was no previous study and it gives the maximum sample size), 95%confidence interval ($Z_{\alpha/2}$ = 1.96), and margin of error (d = 0.05).

$$n = \frac{(Z_{\alpha/2})^2 \mathrm{p}(1-\mathrm{p})}{\mathrm{d}^2} \quad n = \frac{(1.96)^2(0.5)(0.5)}{(.05)2} = 384$$

With the assumption of 10% non- and incomplete responses, the final sample size was calculated to be 422. Convenience sampling was used to select the health facilities and all eligible professionals who were at work at the time of the study.

### Inclusion and exclusion criteria

**Inclusion criteria.** Physicians working at selected public hospitals and pharmacy professionals from the public hospitals, community pharmacies, pharmaceutical industries, EFDA, Ethiopian pharmaceutical supply services, and Ministry of Health and who were willing to participate were included in the study.

**Exclusion criteria.** Physicians and pharmacy professionals who were not willing to participate in the study and not present at the time of data collection were excluded.

### Data collection

Data were collected using a self-administered structured questionnaire (S1 File). The questionnaire was pre-tested on professionals working in a hospital and a pharmaceutical industry which were not included in the final analysis. Information from the pre-test helped to improve the clarity of some questions and the structure of the questionnaire. The study participants were selected from five hospitals, five pharmaceutical industries, eight community pharmacies, the national regulatory authority, the national pharmaceutical supply agency and the ministry of health. The questionnaire was organised in three major sections. The first section included questions on the respondents' socio-demographic characteristics. The second section solicited information on participants' preferences on the source of medicines and their reasons; a list of possible reasons was given for participants to choose from. The third section asked questions about participants' knowledge of BE, their awareness about the presence of a local BE center and its perceived advantages. Respondents were asked to choose one or more options from a list of possible answers for questions related to their perceptions on when products can be called bioequivalent, dosage forms that need BE study and advantages of conducting BE

studies locally. The respondents were identified and approached by the data collectors at work during the time of data collection. Some respondents filled and returned the questionnaire immediately while the majority returned the filled questionnaires a few days later.

## Data analysis

After data clean-up, data entry and statistical analysis was performed using SPSS 25.0. Data were summarized in descriptive statistics. Chi-square test was done to examine the relationship between participants' characteristics and their preference on source of medicines. The options for preferences were categorised into 'locally manufactured products', 'imported products' and 'indifferent'. Logistic regression was performed to evaluate the effect of the independent variables on participants' preference on the source of medicines. Multinomial logistic regression analysis was used to assess factors associated with participants' preference towards the source of medicines. In the multinomial logistic regression analysis, "preference of locally manufactured medicines" was the reference category for the dependent variables, and was compared with the other categories ("preference of imported medicines" versus "preference of locally manufactured medicines"; "indifferent towards the source of medicine" versus "preference towards locally manufactured medicines"). The $p$-value$<0.05$ was considered as statistically significant and both crude (COR) and adjusted odds ratio (AOR) with 95% CI) were reported. Chi-square was performed to test the model and see whether change in unexplained variance from the baseline model to the final model is significant or not.

## Ethical considerations

Approval to conduct this research was obtained from the Ethical Review Board of School of Pharmacy, Addis Ababa University (ERB/SOP/21/10/2018). Before data collection, written informed consent was taken from each participant after they were assured of anonymity and confidentiality of their responses, informed about the purpose of the study and that all participation was voluntary.

## Results

### Participant characteristics

From the total of 422 professionals who consented to take part in the study, 416 returned completed questionnaires making the response rate 98.6%. Most of the participants (272, 65.4%) were male and 184 (44.2%) were between the age of 26 and 30 years with a median age of 29 years (IQR: 26.6, 34). Over three quarters (76.4%) of the participants were pharmacy professionals and the majority (189, 45.4%) had Bachelor degree in pharmacy. With respect to current area of practice, more than half (226, 54.3%) of the respondents worked in hospitals followed by pharmaceutical industries (99, 23.8%) (Table 1).

### Participants' preference on source of medicines

Participants were asked about which products they preferred to recommend, prescribe or dispense to patients and 124 (29.8%) stated their preference for locally manufactured products. A total of 194 (46.6%) and 98 (23.6%) mentioned preferring imported products and having no specific preference, respectively. Most physicians 58(59.2%) preferred imported medicines than the locally manufactured ones when they prescribe. Higher proportion of respondents working in health authorities preferred imported medicines. Chi-square test showed that participants' academic qualification, practice setting and the number of years in their current

**Table 1. Socio-demographic characteristics of health professional participants practicing in Addis Ababa, Ethiopia.**

| Characteristics | n | (%) | Mean ± SD, Median (IQR) |
|---|---|---|---|
| Gender | | | |
| Male | 272 | (65.4) | |
| Female | 144 | (34.6) | |
| Age (in years): | | | 30.9 ± 6.5, 29 (26.6, 34) |
| 20–25 | 70 | (16.8) | |
| 26–30 | 184 | (44.2) | |
| 31–35 | 88 | (21.2) | |
| > 35 | 74 | (17.8) | |
| Highest academic qualification | | | |
| Bachelor Degree and above in Pharmacy | 217 | (52.2) | |
| Diploma in Pharmacy | 101 | (24.3) | |
| Physician | 98 | (23.5) | |
| Practice setting | | | |
| Hospital | 226 | (54.3) | |
| Community pharmacy | 20 | (4.8) | |
| Pharmaceutical industry | 99 | (23.8) | |
| Regulatory authority | 32 | (7.7) | |
| Pharmaceutical supply agency | 29 | (7.0) | |
| Ministry of health | 10 | (2.4) | |
| Number of years in current practice setting: | | | |
| Median (IQR) | | | 5.4 ± 4.7, 4(2, 7) |
| ≤2 | 110 | (26.5) | |
| 3–5 | 164 | (39.4) | |
| > 5 | 142 | (34.1) | |

practice setting were significantly ($p< 0.05$) associated with their preference on the sources of medicines (Table 2).

## Predictive factors for source of medicine preference

To assess the possible factors associated with the preference of healthcare providers towards the source of medicines, all the studied variables were included in the multinomial logistic regression analysis. Hence, age of the participants and sex were excluded and only qualification, practice setting and years of experience were included in the final model.

After adjusting for potential confounders, the variables that were found to be significantly predicted participants' preference towards imported vs locally manufactured medicines were academic qualification and practice setting whereas all the variables included (academic qualification, practice setting and years of experience) in the final model were significantly associated towards indifferent vs local manufactured medicines preference.

The odds of preference towards imported products was reduced by 60% among diploma holders of pharmacy (AOR = 0.40; 95%CI: 0.18–0.91, *p = 0.028*) and 68% among pharmacy bachelor degree and above holders (AOR = 0.32; 95%CI: 0.15–0.68, *p = 0.003*) compared with physicians. Besides, those who practiced in pharmaceutical industries (AOR = 0.40, 95%CI: 0.21–0.77, *p = 0.006*) preferred locally manufactured products as compared to those practicing in the hospital.

Regarding the comparison of indifferent groups vs preference of locally manufactured medicines; the odds of being indifferent towards the source compared with local product

**Table 2. Medicine source preferences and associated factors among health professionals practicing in Addis Ababa, Ethiopia.**

| | | Participants' preference on source of medicines | | | p-value* |
|---|---|---|---|---|---|
| | | Locally manufactured | Imported | Indifferent | |
| | | N(%) | N(%) | N(%) | |
| Overall | | 124(29.8) | 194(46.6) | 98(23.6) | |
| Sex | | | | | 0.697 |
| | Male | 78(28.7) | 127(46.7) | 67(24.6) | |
| | Female | 46(31.9) | 67(46.5) | 31(21.5) | |
| Age | | | | | 0.664 |
| | 20–25 | 24(34.3) | 32(45.7) | 14(20.0) | |
| | 26–30 | 59(32.1) | 84(45.7) | 41(22.3) | |
| | 31–35 | 20(22.7) | 45(51.1) | 23(26.1) | |
| | > 35 | 21(28.4) | 33(44.6) | 20(27.0) | |
| Qualification | | | | | <0.0001 |
| | Physician | 14(14.3) | 58(59.2) | 26(26.5) | |
| | Diploma in pharmacy | 41(40.6) | 49(48.5) | 11(10.9) | |
| | Bachelor degree and above in pharmacy | 69(31.8) | 87(40.1) | 61(28.1) | |
| Practice setting | | | | | <0.0001 |
| | Hospital | 54(23.9) | 112(49.6) | 60(26.5) | |
| | Community pharmacy | 6(30.0) | 9(45.0) | 5(25.0) | |
| | Pharmaceutical industry | 50(50.5) | 31(31.3) | 18(18.2) | |
| | Health authorities** | 14(19.7) | 42(59.2) | 15(21.1) | |
| Number of years in current practice setting | | | | | 0.024 |
| | ≤2 | 31(28.2) | 52(47.3) | 27(24.5) | |
| | 3–5 | 57(34.7) | 81(49.4) | 29(17.7) | |
| | >5 | 36(25.3) | 61(43.0) | 45(31.7) | |

*p-value based on chi-square analysis

**Include Ethiopian Ministry of Health, Ethiopian Food and Drug Authority and Ethiopian Pharmaceutical Supply Services

preference among pharmacy diploma holders was reduced by 76% (AOR = 0.24, 95%CI: 0.09–0.67, *p = 0.006)* as compared with physicians. Similarly, the odds of indifferent towards the source of medicine compared with local product preference found to be lower among those who practiced in the pharmaceutical industry (AOR = 0.35; 95%CI: 0.16–0.75; *p = 0.007*) than those from hospitals. Moreover, the odds of being indifferent towards the source compared with local products among those participants who had working experience from three to five years was reduced by 67% (AOR = 0.33, 95%CI: 0.16–0.72, *p = 0.005)* compared with those who had >5-year experience.

The change in unexplained variance from the baseline model (-2LL = 824.0) to the final model (-2LL = 760.5) was significant (Model $X^2$ = 63.4, *p-value<0.0001*). This change is significant and the final model is a better fit than the original model (Table 3).

## Reasons for source of medicine preference

Easy availability (76, 61.3%) and cheap price (73, 58.9%) were the major reasons mentioned by most of the respondents for preferring locally manufactured medicines. In contrast, better quality (139, 71.6%) and greater effectiveness (106, 54.6%) were the most frequently mentioned reasons for preferring imported pharmaceuticals (Table 4).

**Table 3. Multinomial logistic regression analysis of factors associated with preference on sources of medicines among health professionals practicing in Addis Ababa, Ethiopia.**

| | | | B(SE) | COR (95% CI) | B(SE) | AOR (95% CI) |
|---|---|---|---|---|---|---|
| **Imported Vs Local** | | | | | | |
| | Age in years | | 0.02(0.02) | 1.02(0.98–1.06) | 0.01(0.03) | 1.01(0.96–1.06) |
| | Sex | Female | | 1.00 | | 1.00 |
| | | Male | 0.11(0.24) | 1.12(0.70–1.79) | 0.22(0.27) | 1.25(0.73–2.13) |
| | Qualification | Physician | | 1.00 | | 1.00 |
| | | Diploma in Pharmacy | -1.12(0.37) | 0.29(0.14–0.59)* | -0.91(0.42) | 0.40(0.18–0.91)* |
| | | Bachelor degree and above in pharmacy | -1.19(0.34) | 0.30 (0.16–0.59)** | -1.14(0.38) | 0.32(0.15–0.68)* |
| | Practice setting | Hospital | | 1.00 | | 1.00 |
| | | Community pharmacy | -0.32(0.55) | 0.72(0.25–2.14) | 0.06(0.58) | 1.06(0.34–3.33) |
| | | Pharmaceutical industry | -1.21(0.28) | 0.30(0.17–0.52)** | -0.91(0.33) | 0.40(0.22–0.77)* |
| | | Health authorities | 0.37(0.35) | 1.45(0.73–2.87) | 0.74(0.39) | 2.10(0.98–4.47) |
| | Years of experience | > 5 | | 1.00 | | 1.00 |
| | | 3–5 | -0.18(0.27) | 0.84(0.49–1.43) | -0.12(0.33) | 0.89(0.46–1.71) |
| | | ≤ 2 | -0.01(0.31) | 0.99(0.54–1.82) | -0.11(0.41) | 0.90(0.41–1.99) |
| **Indifferent Vs Local** | | | | | | |
| | Age in years | | 0.03(0.02) | 1.03(0.99–1.08) | -0.01(0.03) | 0.99(0.93–1.05) |
| | Sex | Female | | 1.00 | | 1.00 |
| | | Male | 0.24(0.29) | 1.28(0.73–2.23) | 0.33(0.32) | 1.39(0.74–2.61) |
| | Qualification | Physician | | 1.00 | | 1.00 |
| | | Diploma in Pharmacy | -1.9(0.48) | 0.14 (0.06–0.37)** | -1.44(0.53) | 0.24(0.09–0.67)* |
| | | Bachelor degree and above in pharmacy | -0.74(0.38) | 0.48(0.23–0.99)* | -0.45(0.43) | 0.64(0.28–1.47) |
| | Practice setting | Hospital | | 1.00 | | 1.00 |
| | | Community pharmacy | -0.29(0.63) | 0.75(0.22–2.60) | -0.15(0.69) | 0.86(0.22–3.34) |
| | | Pharmaceutical industry | -1.13(0.33) | 0.32(0.17–0.62)* | -1.06(0.39) | 0.35(0.16–0.75)* |
| | | Health authorities | -0.04(0.42) | 0.96(0.43–2.18) | -0.22(0.46) | 0.81(0.33–1.98) |
| | Years of experience | > 5 | | 1.00 | | 1.00 |
| | | 3–5 | -1.01(0.33) | 0.37(0.19–0.69)* | -1.1(0.39) | 0.33(0.16–0.72)* |
| | | ≤2 | -0.36(0.35) | 0.70(0.35–1.37) | -0.69(0.46) | 0.50(0.20–1.23) |

AOR = adjusted odds ratio, COR = Crude odds ratio

*statistically significant at $p < 0.05$

** statistically significant at $p < 0.001$

**Table 4. Participants' stated reasons for their preferences on source of medicines.**

| Reasons for preference on source of medicines | Participants' preference on source of medicines | | | |
|---|---|---|---|---|
| | Locally manufactured | | Imported | |
| | n* | (%) | n* | (%) |
| Better quality | 35 | (28.2) | 139 | (71.6) |
| Easily available | 76 | (61.3) | 18 | (9.3) |
| Cheap price | 73 | (58.9) | 14 | (7.2) |
| More effective | 21 | (16.9) | 106 | (54.6) |
| Well promoted | 6 | (4.8) | 27 | (13.9) |
| Others** | | | 4 | (2.1) |

*Multiple selections were allowed and hence numbers do not add up to n = 100%

**Others include fast relief, pressure from patients

## Participants' perception about BE and BE studies

Majority of the participants (309, 74.3%) defined BE correctly as 'same drugs with similar safety and efficacy'. The second most frequently (225, 54.1%) stated definition was 'same drugs with same strength'. Most of the respondents (249, 59.9%) believed that BE study should be required for all immediate release oral products for registration by the regulatory body of Ethiopia. Tablet (319, 76.7%) and capsule (292, 70.2%) were the most frequently mentioned dosage forms that require BE study. Moreover, 106 (25.5%) didn't believe that local manufacturers carryout BE studies for products that require it. Among this, the most frequently mentioned reasons for failing to perform BE studies were absence of regulatory requirement (72, 67.9%) and inaccessibility of BE center (51, 48.1%). A total of 321 (77.2%) participants believed that conducting BE study locally has advantages. However, only 139 (33.4%) of the respondents knew about the presence of a regional BE center in the country. Accessibility of the center (230, 71.7%) and affordability (204, 63.6%) were among the major reasons mentioned by respondents as potential benefits.

## Discussion

This study showed the views of physicians from hospitals and pharmacy professionals working in different health sectors towards their preference on the source of medicines and BE studies of generic medicines. Most of the participants (194, 46.6%) stated their preference for imported products while only 124 (29.8%) preferred the locally manufactured ones. This was in line with similar studies that reported physicians and pharmacists being highly concerned about the manufacturing sources of drugs and preferred imported products [21, 22]. The multinomial logistic regression analysis revealed that compared to physicians, participants with diploma (AOR = 0.40; 95%CI: 0.18–0.91, *p = 0.028*) and bachelor and above level (AOR = 0.32; 95%CI: 0.15–0.68, *p = 0.003*) education in pharmacy were more likely to prefer locally manufactured products. This difference could be due to the better awareness and/or understanding of pharmacy professionals about manufacturing of medicines, their quality attributes and approval system for locally produced drugs. This finding is in line to results of other studies in which more than 50% of pharmacy professionals believed that locally manufactured generics have similar safety and efficacy profiles when compared to imported generics [23–25]. Moreover, those participants who practiced in pharmaceutical industries (AOR = 0.40, 95%CI: 0.21–0.77, *p = 0.006*) preferred locally manufactured products as compared to those working in the hospital (Table 3). The confidence of the participants who worked in pharmaceutical industries may have come from their knowledge of the process of production in their respective companies. Even though drug shortage is a global issue, it affects mainly the developing countries like most of African nations due to limited local medicine manufacturing capacity, and are heavily dependent on imported medicines from India and China. The situation in Ethiopia is not an exceptional one and the shortage was intensified especially during the COVID-19 pandemic despite the attention given for the pharmaceutical manufacturing by the government. Establishment of the Kilinto Industrial Park is among the measures taken by the Ethiopian government to expand the production of pharmaceuticals and respond to the unmet demand of medicines in the country [19, 20].

Quality, efficacy, price and availability were among factors influencing the respondents' choice of generic medicine sources. Alemayehu *et al* reported the concern of Ethiopian physicians about the effectiveness of locally produced medicines [16]. This is further supported by the findings of studies done elsewhere [21, 22, 26, 27]. In the current study, higher quality and efficacy were mentioned as justifications by majority of respondents for choosing imported medicines over the domestically produced ones. On the other hand, the easy availability and

lower price were the major reasons suggested by the respondents for their preference of locally manufactured drugs. In a study conducted in Ethiopia, 62.4% of pharmacy professionals claimed locally manufactured generics are cheaper compared to imported generics [23]. Malaysian and Afghan studies also found that domestic generics were cheaper than their imported counterparts [24, 25].

Generic drugs must demonstrate BE to innovator products through appropriate BE studies [28]. To evaluate the respondents' knowledge on BE, they were asked about its definition and potential oral dosage forms with BE requirement. Majority of participants (74.3%) correctly identified the definition of BE for generic products (Fig 1). They stated that bioequivalent products will exhibit essentially similar efficacy and safety profiles. This is the basic assumption in BE between two products [29–31]. However, more than half of the participants (54.1%) didn't clearly know when two products are said to be BE. They stated that two products are said to be BE when they exhibited similarity for identity and strength. Such high number of respondents' misapprehension about BE definition is somewhat alarming.

Oral solids (like tablets and capsules) and oral suspensions are under list of dosage forms that require BE studies for product registration. However, syrups are among dosage forms for which waiver for BE testing is requested since it does not typically involve dissolution process for its absorption and is not related with bioavailability problem [32]. Though majority of respondents had correctly pointed out tablets and capsules (76.7% and 70.2%, respectively) as dosage forms with BE requirement, more than half of the study participants (58.4%) also chose syrups as a dosage form that needs BE study that again showed gap in the know-how of the respondents.

Majority of the participants (310, 74.5%) believed that the local manufacturers implement BE studies for their generic products which indicated a misperception given the fact that locally produced generic drugs in Ethiopia are approved without proof of BE [16, 33]. The most pertinent perceived reason mentioned by participants for local manufacturers not performing BE studies was lack of enforcement by the country's regulatory body (67.9%). This is true that, currently, there is a double standard in enforcing BE requirements by EFDA for imported and locally produced medicines. While imported medicines are subject to strict requirements to present evidence of BE studies, local pharmaceutical manufacturers are exempted from this requirement in clear contradiction of the requirements for BE study as indicated in the EFDA guideline for registration of medicines [32]. It is evident that sub-Saharan African countries have weak regulatory standards and enforcement as compared to global standards [5]. Significant proportion of respondents also raised the issue of inaccessibility of a BE center (48.1%) and high cost of BE studies (41.5%) for their perceived reasons why local manufacturers failed to conduct BE studies for their generics. It is reported elsewhere that local manufacturers in developing countries are facing problems of accessing centres to conduct BE studies [10]. The absence of BE testing facilities in the East African region was mentioned in other studies as one of the reasons for local manufacturers not submitting BE study data for product registration [16, 34]. Clearly, this shows that there is information gap regarding the existence of a regional bioequivalence center (RBEC) which is based in Addis Ababa, Ethiopia.

In addition, 42.5% of respondents indicated poor commitment of companies for not conducting BE studies for their generic products. Obviously, the current era of 'non-competitive' marketplace for local manufacturers is not expected to continue for long. The increasing interest that the international companies are showing to invest in Ethiopia and the advanced technology that they will be introducing into the pharmaceutical manufacturing sector will pose serious challenges to the local manufacturers that are currently comfortable with their current performance. Accordingly, it would do them well to plan strategically and prepare for this

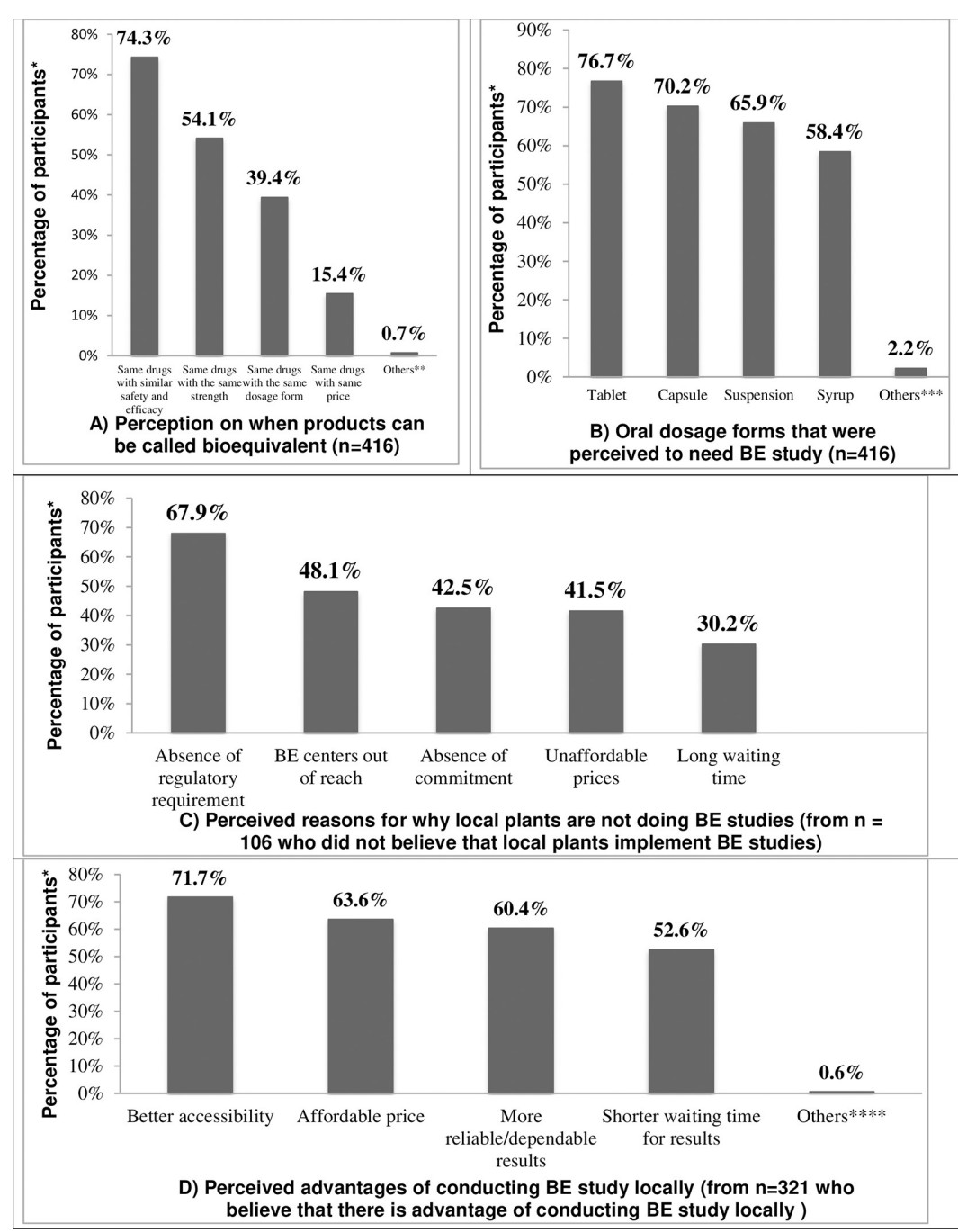

**Fig 1. Respondents' perception on BE studies and the advantages and challenges of doing BE studies locally.** *Percentages do not add up to hundred because participants can select multiple responses. **Other definition: same drug with different price, same drug with different brand. ***Other dosage forms: products unrelated to the immediate release oral dosage forms. ****Other advantages: economic gain for the country.

future. Proof of BE will also benefit companies to reach a wider market place including countries with strict regulatory requirements.

Respondents also suggested the potential benefits of conducting BE studies in the local BE center. The majority of them (66.6%) were not aware of the presence of a BE center in the country which indicates the limitation of promotional activities and communication work by

the center as well as pertinent stakeholders about the center. This might have contributed to the underutilization and lack of support to the center by different organizations. Despite this, many of the study participants (77.2%) agreed that local companies will greatly benefit if BE studies for their generics are conducted in a local BE center. Accessibility (71.7%) of the facility and affordability (63.6%) of the BE service were mentioned by the majority of respondents as potential benefits. A feasibility study conducted in Eastern Africa for setting up BE center in the region showed more than 75% cost saving can be achieved by local companies as a result of conducting BE studies in local centers compared to those done in the United States, Canada, Europe, India and South Africa [35].

Ethiopia has identified the pharmaceutical sector as one of the primary focus areas of development in its growth and transformation plan [19, 20, 36]. In line with this, a quality assured pharmaceutical industrial park was established to develop a pharmaceutical hub for the region [19]. This is expected to attract international companies to invest in Ethiopia where there is still huge unmet market need for pharmaceuticals. Clearly, the attention given by the government to boost the pharmaceutical industry sector should be coupled with strengthening the regulatory framework to ensure the production of quality medicines. Implementing BE studies is one way of ensuring the quality of generic products which requires the availability of a fully functional and accredited BE study facility.

The study has certain limitations that may affect the generalizability of our findings. The first is that the study participants were selected using purposive sampling technique and there might be a selection bias; the second is that the study was done only in Addis Ababa, the capital city of Ethiopia which might affect the generalizability of our findings to all healthcare professionals working in Ethiopia. Causal relationship cannot be drawn between healthcare professional preference regarding the source of medicines and predictors due to the cross-sectional nature of the study. Despite these limitations, this study addresses a timely and relevant question, which is not well researched in Ethiopia. Thus, the findings can be used as input to let manufacturers and regulators devise ways to increase health professionals' confidence in local products, promote the importance of BE study for generic medicines and strengthen local BE study capacity.

## Conclusion and recommendation

The present study revealed that respondents' preference for imported medicine was primarily linked to the belief that they have better quality and efficacy compared to local products. Misconceptions about BE was also observed in some of the respondents including its definition, candidate dosage forms and requirements, which entail appropriate interventions like trainings as continuous professional development program or incorporation of appropriate BE issues in academic curricula. About quarter of the study participants believed that local generics were approved without BE data. Supported by reality, this belief may have negatively affected the trust of prescribers and dispensers on local generic medicines as shown by their preference of imported medicines over locally manufactured ones. Majority of respondents claimed the lack of BE study enforcement had made local companies reluctant to submit BE studies for their generic products' registration. Physicians and pharmacy professionals need assurance on the quality of local products to enhance their confidence in them. Enforcement of BE studies is among the measures to be taken by EFDA which will benefit consumers, healthcare providers, government and the local manufacturers as well. Majority of respondents have acknowledged the benefits of availability of the BE center in the country. Hence the local BE center should be supported, financially and technically, by relevant stakeholders to make it fully operational.

## Supporting information

**S1 File. Data collection tool.**
(DOCX)

## Acknowledgments

The authors would like to thank all physicians and pharmacy professionals who voluntarily participated in this study.

## Author Contributions

**Conceptualization:** Muluken Nigatu Selam, Abrham Temesgen Mengstu, Gebremedhin Beedemariam Gebretekle, Anteneh Belete.

**Data curation:** Muluken Nigatu Selam.

**Formal analysis:** Atalay Mulu Fentie, Eskinder Eshetu Ali.

**Funding acquisition:** Muluken Nigatu Selam, Abrham Temesgen Mengstu, Gebremedhin Beedemariam Gebretekle, Anteneh Belete.

**Investigation:** Muluken Nigatu Selam.

**Methodology:** Muluken Nigatu Selam, Abrham Temesgen Mengstu, Atalay Mulu Fentie, Gebremedhin Beedemariam Gebretekle, Eskinder Eshetu Ali, Anteneh Belete.

**Project administration:** Anteneh Belete.

**Resources:** Muluken Nigatu Selam, Anteneh Belete.

**Supervision:** Anteneh Belete.

**Validation:** Muluken Nigatu Selam, Abrham Temesgen Mengstu, Atalay Mulu Fentie, Gebremedhin Beedemariam Gebretekle, Eskinder Eshetu Ali, Anteneh Belete.

**Visualization:** Muluken Nigatu Selam, Abrham Temesgen Mengstu, Atalay Mulu Fentie, Gebremedhin Beedemariam Gebretekle, Eskinder Eshetu Ali.

**Writing – original draft:** Muluken Nigatu Selam, Abrham Temesgen Mengstu, Gebremedhin Beedemariam Gebretekle, Eskinder Eshetu Ali, Anteneh Belete.

**Writing – review & editing:** Muluken Nigatu Selam, Abrham Temesgen Mengstu, Atalay Mulu Fentie, Gebremedhin Beedemariam Gebretekle, Eskinder Eshetu Ali, Anteneh Belete.

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
