## [Decision Letter · Decision Letter 0]

1 Nov 2022

PONE-D-22-22508

Health professionals’ perceptions on local production and bioequivalence study of generic medicines: A cross-sectional survey of physicians and pharmacy professionals in Addis Ababa, Ethiopia

PLOS ONE

Dear Dr. Belete,

Thank you for submitting your manuscript to PLOS ONE. After careful consideration, we feel that it has merit but does not fully meet PLOS ONE’s publication criteria as it currently stands. Therefore, we invite you to submit a revised version of the manuscript that addresses the points raised during the review process.

We look forward to receiving your revised manuscript.

Kind regards,

Damir Erceg, MD, PhD, Assoc. Prof.

Academic Editor

PLOS ONE

Journal Requirements:

“This research was financially supported by a research grant from the Addis Ababa University.”

4. We note you have included a table to which you do not refer in the text of your manuscript. Please ensure that you refer to Table 2 in your text; if accepted, production will need this reference to link the reader to the Table.

Additional Editor Comments:

There are enclosed suggestions and comments from two reviewers.

Reviewers' comments:

Reviewer's Responses to Questions

**Comments to the Author**

1. Is the manuscript technically sound, and do the data support the conclusions?

Reviewer #1: Yes

Reviewer #2: Yes

2. Has the statistical analysis been performed appropriately and rigorously? 

Reviewer #1: Yes

Reviewer #2: I Don't Know

3. Have the authors made all data underlying the findings in their manuscript fully available?

Reviewer #1: No

Reviewer #2: Yes

4. Is the manuscript presented in an intelligible fashion and written in standard English?

Reviewer #1: Yes

Reviewer #2: Yes

5. Review Comments to the Author

Reviewer #1: Thank you for the invitation to review this manuscript which reports a cross-sectional survey of health professionals’ perceptions on local production and bioequivalence study of generic medicines in Addis Ababa, Ethiopia. Apart from a few typos here and there, the manuscript is well-designed, well-written and have several methodological strengths, and I commend the authors for conducting this work.

The authors mentioned India and China as main sources of medicines source. It’d be good to discuss the implication of the recent supply chain disruption caused by the COVID pandemic, and how the government responded – at the policy and practice level – to mitigate the consequences, both in the short and long run.

Reviewer #2: I recommend adding to the manuscript the content of the self-administered structured questionnaire to get a complete insight into the exact wording of questions, and predefined answers  (if any exist). From the manuscript, it is not clear were the categories mentioned in Table 4 (Better quality, Easily available, Cheap price, More effective and Well promoted) were predefined or concluded from the free-text answer of the participant. It is the same with the "Perception on when products can be called bioequivalent" where also it is not clear were provided categories (Same drugs with similar safety and efficacy, Same drugs with the same strength, Same drugs with the same dosage form and Same drugs with the same price) were predefined or concluded from the free-text answer of the participant.

The link in reference No. 4 (line 380) should be updated as a provided hyperlink in the pdf of the manuscript is not functioning. The reference is not accessible. It is the same with the links in references: No. 19 (line 418/419); No. 20 (line 424) and No. 32 (line 455).

6. PLOS authors have the option to publish the peer review history of their article (what does this mean?). If published, this will include your full peer review and any attached files.

Reviewer #1: No

Reviewer #2: **Yes: **Csaba Dohoczky

---

## [Author Response · Author response to Decision Letter 0]

19 Dec 2022

To

Academic Editors and Reviewers

The PLOSE ONE

PONE-D-22-22508

Subject: Responses to the academic editor and the reviewers’ comments on: “Health professionals’ perceptions on local production and bioequivalence study of generic medicines: A cross-sectional survey of physicians and pharmacy professionals in Addis Ababa, Ethiopia”

Dear Editor, Assoc Prof. Damir Erceg 

First, we would like to thank academic editor and the reviewers for the constructive comments and suggestions that significantly improved our work. We have done our best to address your questions and concerns, and we have made the corresponding changes in the manuscript. 

Our point-by-point responses to the comments are presented below. The comments given by the reviewers are indicated on the left and point-by-point responses to reviewers on the right of the table and indicated with track changes in the revised manuscript. Regarding a data availability statement, the data collection tool is attached as a separate supporting information file (S1 file).

The changes and corrections asked by “Editor in Chief”:

Authors’ response.

Here is the point-by-point response to editor and reviewers' comments

Authors’ response.

All the changes to the revised manuscript are indicated by using track changes and submitted as a separate file.

Authors’ response.

Unmarked version of the revised paper is submitted as a separate file

Authors’ response.

Now, the revised manuscript complies with the format of the journal and files are named accordingly.

Authors’ response.

The role of the funder is indicated in the financial disclosure of the revised manuscript as suggested.

6. Upon re-submitting your revised manuscript, please upload your study’s minimal underlying data set as either Supporting Information files or to a stable, public repository and include the relevant URLs, DOIs, or accession numbers within your revised cover letter.

Authors’ response.

The minimal underlying data set are already indicated in the manuscript and the statement is corrected accordingly in the revised manuscript. Moreover, the data collection tool is submitted as a supporting information file.

7. We note you have included a table to which you do not refer in the text of your manuscript. Please ensure that you refer to Table 2 in your text; if accepted, production will need this reference to link the reader to the Table.

Authors’ response.

The revised manuscript mentioned Table 2 in the text before the appearance of the table.

8. Please review your reference list to ensure that it is complete and correct.

Authors’ response.

The list of references is reviewed and revised to make it complete and correct and comply the journal requirements.

The comments asked by the reviewers and corrections made by the authors 

Reviewer comments Response

1 Have the authors made all data underlying the findings in their manuscript fully available? 

Reviewer 1: No

Reviewer 1: Yes - The statistics of the study is indicated in the manuscript. However, the data collection tool was missed in the previously submitted manuscript and now included in the revised manuscript as a separate file.

2 Reviewer #1: Thank you for the invitation to review this manuscript which reports a cross-sectional survey of health professionals’ perceptions on local production and bioequivalence study of generic medicines in Addis Ababa, Ethiopia. Apart from a few typos here and there, the manuscript is well-designed, well-written and have several methodological strengths, and I commend the authors for conducting this work. - Thank you and the English language editing is made by all authors on the revised manuscript and indicated as track changes. 

3 Reviewer #1: The authors mentioned India and China as main sources of medicines source. It’d be good to discuss the implication of the recent supply chain disruption caused by the COVID pandemic, and how the government responded – at the policy and practice level – to mitigate the consequences, both in the short and long run. - The impact of dependency on imported medicine and the attention given by the government is indicated in the discussion section of the revised manuscript (line 260-267).

4 Reviewer #2: I recommend adding to the manuscript the content of the self-administered structured questionnaire to get a complete insight into the exact wording of questions, and predefined answers (if any exist). From the manuscript, it is not clear were the categories mentioned in Table 4 (Better quality, Easily available, Cheap price, More effective and Well promoted) were predefined or concluded from the free-text answer of the participant. It is the same with the "Perception on when products can be called bioequivalent" where also it is not clear were provided categories (Same drugs with similar safety and efficacy, Same drugs with the same strength, Same drugs with the same dosage form and Same drugs with the same price) were predefined or concluded from the free-text answer of the participant. - Thank you and the data collection tool is provided as a supporting file in the revised manuscript (S1 file).

5 Reviewer #2: In The link in reference No. 4 (line 380) should be updated as a provided hyperlink in the pdf of the manuscript is not functioning. The reference is not accessible. It is the same with the links in references: No. 19 (line 418/419); No. 20 (line 424) and No. 32 (line 455). - Thank you and all references are checked again and corrected to make them accessible

---

## [Decision Letter · Decision Letter 1]

30 Jan 2023

Health professionals’ perceptions on local production and bioequivalence study of generic medicines: A cross-sectional survey of physicians and pharmacy professionals in Addis Ababa, Ethiopia

PONE-D-22-22508R1

Dear Dr. Belete,

We’re pleased to inform you that your manuscript has been judged scientifically suitable for publication and will be formally accepted for publication once it meets all outstanding technical requirements.

Kind regards,

Damir Erceg, MD, PhD, Assoc. Prof.

Academic Editor

PLOS ONE

Additional Editor Comments (optional):

The second review of one reviewer is enclosed. The second review of other reviewer is missing. There was only few minor comments in the first review.

Reviewers' comments:

Reviewer's Responses to Questions

**Comments to the Author**

1. If the authors have adequately addressed your comments raised in a previous round of review and you feel that this manuscript is now acceptable for publication, you may indicate that here to bypass the “Comments to the Author” section, enter your conflict of interest statement in the “Confidential to Editor” section, and submit your "Accept" recommendation.

Reviewer #2: All comments have been addressed

2. Is the manuscript technically sound, and do the data support the conclusions?

Reviewer #2: Yes

3. Has the statistical analysis been performed appropriately and rigorously? 

Reviewer #2: I Don't Know

4. Have the authors made all data underlying the findings in their manuscript fully available?

Reviewer #2: Yes

5. Is the manuscript presented in an intelligible fashion and written in standard English?

Reviewer #2: Yes

6. Review Comments to the Author

Reviewer #2: There are no more comments on the reviewed manuscript to add. The requested questionnaire has been added to the manuscript, and the hyperlinks in the references are corrected.

7. PLOS authors have the option to publish the peer review history of their article (what does this mean?). If published, this will include your full peer review and any attached files.

Reviewer #2: **Yes: **Csaba Dohoczky

---

## [Editor Report · Acceptance letter]

20 Mar 2023

PONE-D-22-22508R1 

Health professionals’ perceptions on local production and bioequivalence study of generic medicines: A cross-sectional survey of physicians and pharmacy professionals in Addis Ababa, Ethiopia 

Dear Dr. Belete:

I'm pleased to inform you that your manuscript has been deemed suitable for publication in PLOS ONE. Congratulations! Your manuscript is now with our production department. 

Kind regards, 

on behalf of

Dr. Damir Erceg 

Academic Editor

PLOS ONE